# LABEL CLEANING WITH A LIKELIHOOD RATIO TEST

## ABSTRACT

To collect large scale annotated data, it is inevitable to introduce label noise, i.e., incorrect class labels. A major challenge is to develop robust deep learning models that achieve high test performance despite training set label noise. We introduce a novel approach that directly cleans labels in order to train a high quality model. Our method leverages statistical principles to correct data labels and has a theoretical guarantee of the correctness. In particular, we use a likelihood ratio test (LRT) to flip the labels of training data. We prove that our LRT label correction algorithm is guaranteed to flip the label so it is consistent with the true Bayesian optimal classifier with high probability. We incorporate our label correction algorithm into the training of deep neural networks and train models that achieve superior testing performance on multiple public datasets.

## 1 INTRODUCTION

Label noise is ubiquitous in real world data. It may be caused by unintentional mistakes of manual or automatic annotators (Yan et al., 2014; Veit et al., 2017). It may also be introduced by malicious attackers (Steinhardt et al., 2017). Noisy labels impair the performance of a model (Smyth et al., 1994; Brodley & Friedl, 1999), especially a deep neural network, which tends to have strong memorization power (Frnay & Verleysen, 2014; Zhang et al., 2017). Improving the robustness of a model to label noise is a crucial yet challenging task in many applications (Mnih & Hinton, 2012; Wu et al., 2018). Existing methods mainly follow two directions, probabilistic reasoning and data selecting.

Probabilistic methods explicitly model a noise *transition matrix*, namely, the probability of one label being corrupted into another (Goldberger & Ben-Reuven, 2017; Patrini et al., 2017). The transition matrix is often estimated from the data, and is used to re-calibrate the training loss or to correct the prediction. Explicit estimation of the transition matrix can be problematic due to the large variation of noise patterns, e.g., uniform noise, asymmetric noise, or mixtures. Furthermore, the transition matrix size is quadratic to the number of classes, making the estimation task prohibitive when the data has hundreds or even thousands of classes.

Data-selecting methods are agnostic of the underlying noise pattern. These methods gradually collect *clean data* whose labels are trustworthy (Malach & Shalev-Shwartz, 2017; Jiang et al., 2018; Han et al., 2018). As more clean data are collected, the quality of the trained models improves. The major issue of these methods is the lack of a quantitative control of the quality of the collected clean data. Without a principled guideline, it is hard to find the correct data collection pace. An aggressive selection can unknowingly accumulate irreversible errors. On the other hand, an overly-conservative strategy can be very slow in training, or stops with insufficient clean data and mediocre models.

We propose a novel method with the benefit from both the probabilistic and the data-selecting approaches. Similar to data-selecting methods, our method continuously improves the purity of the data labels by correcting the noise-corrupted ones. Meanwhile, we improve the classifier using the updated labels. Our label correction algorithm is based on statistical principles and is theoretically guaranteed to deliver a high quality label set. Instead of explicitly estimating the transition matrix, the correction algorithm only depends on the prediction of the current model, denoted as $f$. Using an $f$-based likelihood ratio test, we determine whether the current label of each data should be corrected. Our main theorem proves that the label correction algorithm will clean a majority of noisy labels with high probability.

In practice, we incorporate the label correction algorithm into the training of deep neural networks. Our method iteratively updates the labels of the data while continuously training a deep neural

network. To ensure the deep neural network does not overfit with noise labels that are yet to be corrected, we introduce a new *retroactive loss* term that regulates the model by enforcing its consistency with models in previous epochs. The rationale is that the model in an earlier training stage tends to fit the true signal rather than noise, although its overall performance is sub-optimal. Through experiments on various datasets with various noise patterns and levels, we show that our method produces robust neural network models with superior performance.

To the best of our knowledge, *our method is the first to correct labels with theoretical guarantees*. It is has advantages over both probabilistic methods and data-selecting methods. Compared with other data-selecting methods, it has a better quantitative control of the label quality and thus is less brittle when generalizing to different datasets and different noise patterns. Also note that we are not selecting clean data. Instead, we correct labels and always use the whole training set to train. This brings an additional advantage of fully leveraging the data. Compared with other probabilistic methods, our correction algorithm assumes a rather general family of underlying noise patterns and avoids an explicit estimation of the transition matrix.

## 1.1 RELATED WORK

One representative strategy for handling label noise is to model and employ noise transition matrix to correct the loss. For example, Patrini et al. (2017) propose to correct the loss function with estimated noise pattern. The resulting loss is an unbiased estimator of the ground truth loss, and enables the trained model to achieve better performance. However, such an estimator relies on strong assumptions and could be inaccurate in certain scenarios. Reed et al. (2014) consider modeling the noise pattern with a hidden layer. The learning of this hidden layer is regularized with a feature reconstruction loss, yet without a guarantee that the true label distribution is learned. Another method mentioned in their work is to minimize the entropy of neural network output; however, this method tends to predict a single class. To address this weakness, Hendrycks et al. (2019) propose to utilize a small number of kosher data to pre-train a network and estimate the noise pattern. However, such clean data may not always be available in practice.

Alternatively, another direction proposes to design models that are intrinsically robust to noisy data. Crammer et al. (2009) introduce a regularized confidence weighting learning algorithm (AROW), where parameters of classifiers are assumed normally distributed and the mean and covariance of this distribution is updated during training. The idea here is to preserve the weight distribution as much as possible while requiring the model to maintain predictive ability. The follow-up work (Crammer & Lee 2010) proposes to improve this algorithm by herding the updating direction via specific velocity field (NHERD), achieving better performance. Both of these works impose parametric constraint on parameters, which could prevent classifiers from adapting to complex dataset. Arpit et al. (2017) show that deep neural networks tend to learn meaningful patterns before they over-fit to noisy ones. Based on this observation, they propose to add Gaussian or adversarial noise to input when training with noisy labels, and empirically show that such data perturbation is able to make the resulting model more robust. Other commonly adopted techniques, such as weight decay and dropout, are also shown to be effective in increasing the robustness of trained classifier (Arpit et al. 2017; Zhang et al. 2017). However, the intrinsic reasons for this phenomenon still remains unclear and overfitting to noisy label is still inevitable.

Apart from the above mentioned strategies, one recent work proposes to correct the corrupted labels during training. In particular, Tanaka et al. (2018) propose to jointly train the deep network and estimate the underlying true labels. While achieving improved performance, their method largely relies on the prior distribution and is difficult to deploy under cases where there is a large number of classes. In addition, there are a few other works attempting to select clean data while eliminating noisy ones during training (Malach & Shalev-Shwartz, 2017; Jiang et al., 2018; Han et al., 2018). These methods demonstrate promising results, but lack quantitative control of the quality of the collected clean data.

Finally, beyond deep learning framework, there are several theoretic works that demonstrate the robustness of a variety of losses to label noise (Long & Servedio 2010; Natarajan et al. 2013; Ghosh et al. 2015; van Rooyen et al. 2015). Following the work of (Wang & Chaudhuri 2018), Gao et al. (2016) propose an algorithm that can converge to the Bayesian optimal classifier under different noisy settings. Moreover, they also provide in-depth discussion regarding the performance of k-

nearest neighbor (KNN) classifiers. However, the problem with KNN is that it is computationally intensive and difficult to be incorporated into a learning context. Within the framework of deep learning, there are more efforts that need to be made to bridge theory and practice.

## 2 METHOD

Our method has two synchronized modules, the training module and the label correction module. The training module continues to learn a classifier based on the current labels. Meanwhile, the label correction module uses the prediction of the classifier to correct labels.

We start with some preliminaries necessary for the exposition (Section 2.1). In Section 2.2, we explain our correction algorithm. It uses the prediction of the classifier (trained on noisy labels) for a likelihood ratio test. Based on the test result, it decides whether to correct the label of a data. Theoretically, we prove that under certain assumptions of the prediction, $f$, the algorithm will change the labels to the correct ones, i.e., the ones consistent with the Bayes optimal classifier (Theorem 1).

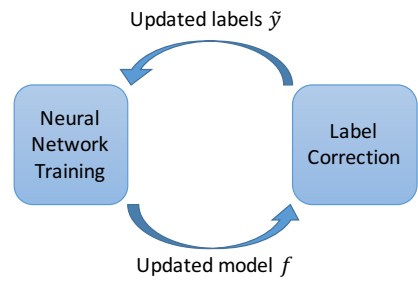

Figure 1: The overview of our method.

In Section 2.3, we present the overall training method. We incrementally train a deep neural network based on the corrected labels, $\widetilde{y}$. Meanwhile, the network's prediction is used for label correction. To improve the quality of the prediction, we introduce a new loss term, called the *retroactive loss*. The goal is to regulate the model using models trained in earlier epochs, as they may be less overfitting with corrupted labels.

For the label correction module, we focus on a binary classifier. But the algorithm and the theoretical results can easily be generalized to the multiclass setting (Corollary 1).

### 2.1 PRELIMINARIES

Let $\mathcal{X}$ be the feature space, $\mathcal{Y} = \{0, 1\}$ be the label space, and $D$ be an unknown distribution on $\mathcal{X} \times \mathcal{Y}$. The joint probability can be factored as $D(\boldsymbol{x}, y) = \Pr(y|\boldsymbol{x})\Pr(\boldsymbol{x})$. We denote by $\eta(\boldsymbol{x}) = \Pr(y = 1|\boldsymbol{x})$ the *true conditional probability*. The *Bayes risk* of a classifier $h : \mathcal{X} \to \mathcal{Y}$ is $R(D, h) = \Pr_{(\boldsymbol{x},y)\sim D}(h(\boldsymbol{x}) \neq y)$. A *Bayes optimal classifier* is the minimizer of the Bayes risk, i.e., $h^* = \arg\min_h R(D, h)$. It can be calculated using the true conditional probability, $\eta$,

$$h^*(\boldsymbol{x}) = \mathbf{1}_{\left\{\eta(\boldsymbol{x})\geq\frac{1}{2}\right\}}(\mathbf{x}) = \begin{cases} 1 & \text{if } \eta(\boldsymbol{x}) >= \frac{1}{2} \\ 0 & \text{otherwise} \end{cases}. \tag{1}$$

We assume the true conditional probability, $\eta$, satisfies the Tsybakov condition (also called the TNC condition) (Tsybakov et al., 2004). This condition, also called margin assumption, stipulates that the uncertainty of $\eta$ (region close to decision boundary at $1/2$) and thus the Bayes optimal classifier is bounded.

**Definition 1** (Tsybakov Condition). *There exist $C > 0, \lambda > 0$, and $t_0 \in (0, \frac{1}{2}]$, such that for all $t \leq t_0$,*

$$\Pr\left[\left|\eta(\boldsymbol{x}) - \frac{1}{2}\right| < t\right] \leq Ct^\lambda.$$

**The noisy label setting.** Instead of samples from $D$, we are given a sample set with noisy labels $S = \{(\boldsymbol{x}, \widetilde{y})\}$ where $\widetilde{y}$ is the possibly corrupted label based on the true label $y$. We assume a *transition probability* $\tau_{i\to j} = \Pr(\widetilde{y} = j|y = i)$, i.e., the chance a ground truth label $y$ is flipped from class $i$ to class $j$. For simplicity, we denote $\tau_{ij} = \tau_{i\to j}$. The transition probabilities $\tau_{01}$ and $\tau_{10}$ are independent of the true joint distribution $D$ and the feature $\boldsymbol{x}$. We denote the conditional probability of the noisy labels as $\widetilde{\eta}(\boldsymbol{x}) = \Pr(\widetilde{y} = 1|\boldsymbol{x})$. In short, we call $\widetilde{\eta}$ the *noisy conditional probability*. It is related linearly to the true conditional probability, $\eta$:

$$\widetilde{\eta}(\boldsymbol{x}) = (1 - \tau_{10})\eta(\boldsymbol{x}) + \tau_{01}[1 - \eta(\boldsymbol{x})] = (1 - \tau_{01} - \tau_{10})\eta(\boldsymbol{x}) + \tau_{01}. \tag{2}$$

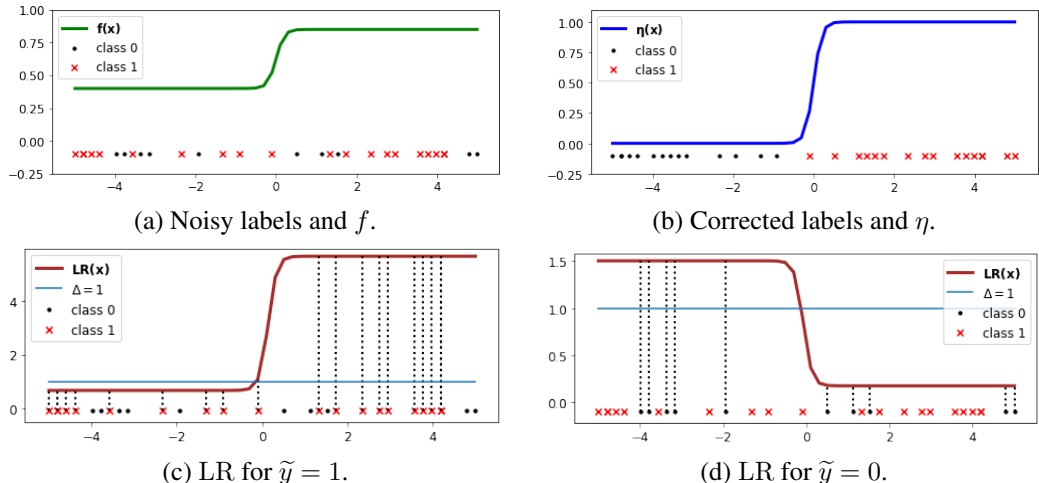

Figure 2: An illustration of the label correction algorithm. $\Delta$ is set to 1. (a): a corrupted sample and its corresponding classifier prediction $f$. (b): after correction, the labels are consistent with the true conditional probability, $\eta$. (c): likelihood ratio for $\widetilde{y} = 1$. Data with $x < 0$ are corrected to $\widetilde{\eta}_{new} = 0$ as $LR(\boldsymbol{x})$ are below $\Delta = 1$. (d): likelihood ratio for $\widetilde{y} = 0$. Data with $x > 0$ are corrected to $\widetilde{\eta}_{new} = 1$ as $LR(\boldsymbol{x})$ are below $\Delta = 1$.

## 2.2 THE LABEL CORRECTION ALGORITHM AND ITS THEORETICAL GUARANTEES

Our label correction algorithm takes in a current neural network prediction $f : \mathcal{X} \to [0, 1]$ (i.e., an estimation of $\eta$ based on the noisy labels). For all training data and their current noisy label $(\boldsymbol{x}, \widetilde{y})$, the correction algorithm uses $f$ to run a likelihood ratio test and to decide whether to flip the label according to the result. The goal of the likelihood test is to decide whether the null hypothesis, $H_0 : \widetilde{y} = y$, is true. If yes, $\widetilde{y}$ is accepted as it is. Otherwise, we flip $\widetilde{y}$, so that hopefully it becomes $y$. Formally, the likelihood ratio is defined as

$$\text{LR}(f, \boldsymbol{x}, \widetilde{y}) = \frac{f(\boldsymbol{x})^{\widetilde{y}} \left[1 - f(\boldsymbol{x})\right]^{1-\widetilde{y}}}{f(\boldsymbol{x})^{1-\widetilde{y}} \left[1 - f(\boldsymbol{x})\right]^{\widetilde{y}}}. \tag{3}$$

We compare this likelihood ratio with a predetermined value $\Delta$. If $\text{LR}(f, \boldsymbol{x}, y) \leq \Delta$, we reject the null hypothesis and flip the label $\widetilde{y}_{new} = 1 - \widetilde{y}$. Otherwise, the label remains unchanged, $\widetilde{y}_{new} = \widetilde{y}$. See Figure 2 for an illustration of the algorithm.

Note that the constant $\Delta$ depends on the underlying noise pattern and $f$. Below we show that if we choose $\Delta$ carefully, the label correction algorithm is guaranteed to make proper correction and clean most of the corrupted labels. However, in practice, $\Delta$ is unknown and needs to be tuned.

**Intuition.** In the likelihood ratio (Eq. (3)), the numerator is the likelihood that the prediction $f$ is consistent with the noisy label $\widetilde{y}$. The denominator is the likelihood of the opposite case. When this ratio is smaller than 1, we know that the prediction of $f$ is more likely to be inconsistent with $\widetilde{y}$. But whether $f$ agrees with $\widetilde{y}$ is not the hypothesis to test. To test the intended null hypothesis ($\widetilde{y} = y$), we need to check whether $\widetilde{y}$ is consistent with the true conditional distribution $\eta$, namely, the Bayes optimal classifier prediction $h^*(\boldsymbol{x})$. To this end, we assume $f$ is a close enough approximate of $\widetilde{\eta}$ as it is trained on the noisy labels. This way, testing whether $f$ agrees with $\widetilde{y}$ is close to testing whether $\widetilde{\eta}$ agrees with $\widetilde{y}$, except that the threshold $\Delta$ needs to be carefully chosen. Another issue we need to consider is that the $\Delta$ is unknown. Our main theorem will bound the chance of failed correction by how close $f$ approximates $\widetilde{\eta}$ and how close we can set $\Delta$ to the perfect one.

### 2.2.1 FORMAL STATEMENT OF THE ALGORITHM AND THE THEOREM

Assume the knowledge of some prediction function $f$ (perhaps from a neural network). The label correction algorithm is given in Procedure 1. It checks the likelihood ratio with regard to a parameter $\Delta = \min(\frac{1 + \tau_{01} - \tau_{10}}{1 + \tau_{10} - \tau_{01}}, \frac{1 + \tau_{10} - \tau_{01}}{1 + \tau_{01} - \tau_{10}})$. If LR is no greater than $\Delta$, we flip the label. Otherwise, $\widetilde{y}_{new}$ is

the same as $\widetilde{y}$. Note that one only needs to know the difference $\tau_{01} - \tau_{10}$ in order to know $\Delta$. In practice, $\Delta$ may be unknown and needs to be decided empirically.

---

**Procedure 1** `LRT-Correction`
**Input:** $(\boldsymbol{x}, \widetilde{y}), f(\boldsymbol{x}), \Delta$.
**Output:** $\widetilde{y}_{new}$
1: **if** $\text{LR}(f, \boldsymbol{x}, \widetilde{y}) \le \Delta$ **then**
2: $\quad \widetilde{y}_{new} = 1 - \widetilde{y}$
3: **else**
4: $\quad \widetilde{y}_{new} = \widetilde{y}$
5: **end if**

**Procedure 2** `LRT-MultiCorrection`
**Input:** $(\boldsymbol{x}, \widetilde{y}), f(\boldsymbol{x}), \Delta$.
**Output:** $\widetilde{y}_{new}$
$\quad y* := \arg\max_i [f(\boldsymbol{x})]_i; \text{LR}(f, \boldsymbol{x}, \widetilde{y}) := \frac{[f(\boldsymbol{x})]_{\widetilde{y}}}{[f(\boldsymbol{x})]_{y*}};$
1: **if** $\text{LR}(f, \boldsymbol{x}, \widetilde{y}) \le \Delta$ **then**
2: $\quad \widetilde{y}_{new} = y*$
3: **else**
4: $\quad \widetilde{y}_{new} = \widetilde{y}$
5: **end if**

---

Our main theorem states that suppose the classifier prediction, $f$, is a close approximation of the noisy conditional probability, $\widetilde{\eta}$. And suppose we can find a good enough $\Delta'$ that is close enough to the ideal $\Delta$. Then there is a very good chance that our algorithm corrects most labels to the correct ones, i.e., the same as the Bayes optimal classifier prediction. Please note that here "proper correction" means that the new label, $\widetilde{y}_{new}$, is the same as the Bayes optimal classifier prediction, $h^*(\boldsymbol{x})$, instead of $y$. This is well justified as it means that the correction will give us a classifier as good as the Bayes optimal one.

**Theorem 1.** *Assume $\eta$ satisfies the Tsybakov condition with constants $C > 0$ and $\lambda > 0$. Let $h^*$ denote the Bayes optimal classifier, and $\widetilde{\eta}$ the noisy conditional probability $\widetilde{\eta}(\boldsymbol{x}) = (1 - \tau_{01} - \tau_{10})\eta(\boldsymbol{x}) + \tau_{01}$. Let $f$ be any classifier such that $||f - \widetilde{\eta}||_\infty \le \epsilon$ for some $\epsilon > 0$. Let $\Delta = \min(\frac{1+\tau_{01}-\tau_{10}}{1+\tau_{10}-\tau_{01}}, \frac{1+\tau_{10}-\tau_{01}}{1+\tau_{01}-\tau_{10}})$, and let $\Delta' > 0$ be a constant such that $\Delta' \in [\Delta - \epsilon, \Delta + \epsilon]$. If $\widetilde{y}_{new}$ denotes the output of the `LRT-Correction` with $\widetilde{y}$, $\boldsymbol{x}$, $f$, and the given $\Delta'$, then*

$$\Pr_{(x,y)\sim D}(\widetilde{y}_{new} \ne h^*) \le C \left( \left| \frac{\tau_{10} - \tau_{01}}{2(1 - \tau_{10} - \tau_{01})} \right| + 2\epsilon \right)^\lambda .$$

*If $\tau_{01} = \tau_{10}$, then $\Pr_{(x,y)\sim D}(\widetilde{y}_{new} \ne h^*) = C (2\epsilon)^\lambda$.*

**Intuition of the proof.** We will prove two lemmas. The first lemma shows that when $f$ is linear in $\eta$, i.e., $f(\boldsymbol{x}) = a\eta(\boldsymbol{x}) + b$, with known coefficients $a, b$, we can find a $\Delta$ such that then the correction algorithm can be correct everywhere. The second lemma proves a more relaxed version. If $f$ is exactly $\widetilde{\eta}$ and if we only know the difference between the transition probabilities, $\tau_{01} - \tau_{10}$, then we can bound the chance of mistakes of the correction algorithm. Finally, based on these two lemmas, careful case analysis, and the Tsybakov condition of the true conditional probability, $\eta$, we can prove the theorem. The complete proof can be found in Appendix A.

**Multiclass setting.** So far, all the description and theoretical results are based on a binary classification setting. However, the results can be generalized to a multiclass setting without any technical difficulties. Let $\widetilde{y}$ be the observed (possibly) corrupted label, $f(\boldsymbol{x})$ be a classifier vector, where $[f(\boldsymbol{x})]_i$ (the $i$th coordinate of $f(\boldsymbol{x})$) specifies the confidence of assigning $\boldsymbol{x}$ to class $i$. The algorithm is presented in Procedure 2. Informally, we present the theoretical guarantee for multiclass LRT as Corollary 1. The proof is left for a journal version.

**Corollary 1.** *Theorem 1 can be generalized to the multiclass setting, when using algorithm* `LRT-MultiCorrection`.

## 2.3 TRAINING DEEP NEURAL NETWORKS WITH LRT-LABEL-CORRECTION

Our training algorithm continuously trains a deep neural network while correcting the noisy labels. Procedure 3 is the pseudocode of the training method, called `AdaCorr`. It trains a neural network model iteratively. Each iteration includes both label correction and model training steps. In label correction step, the prediction of the current neural network, $f$, is used to run LRT test on all training data, and to correct their labels according to the test result. Since $f$ is used to approximate the conditional probability $\widetilde{\eta}$, we use the softmax layer output of the neural network as $f$. After the

labels of all training data are updated, we use them to train the neural network incrementally. We continue this iterative procedure until the whole training converges.

We also have a burn-in stage in which we train the network using the original noisy labels for $m$ epochs. During the burn-in stage, we use the original cross-entropy loss, $L_{CE}$. Afterwards, we add an additional retroactive loss which will be explained below.

**Training with retroactive loss.** After the burn-in stage, we want to avoid overfitting of the neural network, so that its output better approximates $\widetilde{\eta}$. To achieve this goal, we introduce a *retroactive loss* term $L_{retro}(f(\boldsymbol{x}), \widetilde{y})$. The idea is to enforce the consistency between $f$ and the prediction of the model at a previous epoch, $f'$. It has been observed that a neural network at earlier training stage tends to learn the true pattern rather than to overfit the noise (Arpit et al., 2017). Formally, the loss can be written as $\sum_{c=1}^{N_c} f'_c(\boldsymbol{x}) \log f_c(\boldsymbol{x})$,

---

**Procedure 3** `AdaCorr`

**Input:** $S = \{\boldsymbol{x}, \widetilde{y}\}, \Delta, m, T$
 1: **for** epoch=1 to $m$ **do**
 2:     Train neural network with $L_{CE}$
 3: **end for**
 4: $f' = $ current model prediction
 5: **for** epoch=$m + 1$ to $T$ **do**
 6:     **if** epoch $\geq m + 10$ **then**
 7:         $f = $ current model prediction
 8:         **for all** $(\boldsymbol{x}, \widetilde{y}) \in S$ **do**
 9:             $\widetilde{y}_{new}$= `LRT-Correction`($f$,($\boldsymbol{x},\widetilde{y}$),$\Delta$)
10:             $\widetilde{y} = \widetilde{y}_{new}$
11:         **end for**
12:     **end if**
13:     Train using $L_{retro} + L_{CE}$, with $f'$ and $\widetilde{y}$
14: **end for**

---

in which $N_c$ is the number of possible label classes. The training loss is the sum of the retroactive loss and the cross-entropy loss

$$L(f(\boldsymbol{x}), \widetilde{y}) = L_{retro}(f(\boldsymbol{x}), \widetilde{y}) + L_{CE}(f(\boldsymbol{x}), \widetilde{y}) = \sum_{c=1}^{N_c} f'_c(\boldsymbol{x}) \log f_c(\boldsymbol{x}) + \sum_{c=1}^{N_c} \widetilde{y}_c \log f_c(\boldsymbol{x}).$$

In practice, we set $f'$ to be the prediction of the model at the $m$-th epoch. In other words, once the burn-in stage is finished, the training switches from $L_{CE}$ to $L_{CE} + L_{retro}$. And the model at the end of the burn-in stage is used for the retroactive loss. We also set the label correction to start slightly after the burn-in stage, say $m + 10$. The key hyperparameter is the starting epoch $m$. Another hyperparameter is $\Delta$. We select both $m$ and $\Delta$ empirically. Ablation study in Section 4 shows that our method is robust to these hyperparameters.

## 3 EXPERIMENTS

In this section we empirically evaluate our proposed method with several datasets, where noisy labels are injected according to specified noise transition matrices.

**Datasets.** We use the following datasets: MNIST (LeCun & Cortes 2010), CIFAR10 (Krizhevsky et al. a), CIFAR100 (Krizhevsky et al. b) and ModelNet40 (Z. Wu & Xiao 2015). MNIST consists of $28 \times 28$ grayscale images with 10 categories. It contains 60,000 images, and we use 45,000 for training, 5,000 for validation and 10,000 for testing. CIFAR10 and CIFAR100 share the same 60,000 $32 \times 32 \times 3$ image data, with CIFAR10 having 10 categories while CIFAR100 having 100 categories. Similar to MNIST, we split 90% and 10% data from the official training set for the training and validation respectively, and use the official test set for testing. ModelNet40 contains 12,311 CAD models from 40 categories, where 8,859 are used for training, 984 for validation and the remaining 2,468 for testing. We follow the protocol of (Qi et al. (2017)) to convert the CAD models into point clouds by uniformly sampling 1,024 points from the triangular mesh and normalizing them within a unit ball. In all experiments, we use early stopping on validation set to tune hyperparameters and to report the performance on test set.

**Noise patterns.** Following (Reed et al. 2014; Patrini et al. 2017), we corrupt our labels artificially using a noise transition matrix $T$, where $T_{ij} = \tau_{ij} = \Pr(\widetilde{y} = j | y = i)$ is the probability that category $i$ is flipped to category $j$. In our work we focus on two types of $T$: (1) uniform, where the true label $i$ is flipped to other classes with equal probabilities, i.e., $T_{ij} = p/(N_c - 1)$ for $i \neq j$ and $T_{ii} = 1 - p$, where $p$ is the noise level and $N_c$ is the class number; (2) pair flipping, where the

true label $i$ is flipped to $j$ with $T_{ij} = p$ for $i \neq j$ and $T_{ii} = 1 - p$. Examples of these two types of transition matrices are as follows:

$$
T_1 = \begin{pmatrix} 0.7 & 0.1 & 0.1 & 0.1 \\ 0.1 & 0.7 & 0.1 & 0.1 \\ 0.1 & 0.1 & 0.7 & 0.1 \\ 0.1 & 0.1 & 0.1 & 0.7 \end{pmatrix} \qquad T_2 = \begin{pmatrix} 0.7 & 0.3 & 0.0 & 0.0 \\ 0.0 & 0.7 & 0.3 & 0.0 \\ 0.0 & 0.0 & 0.7 & 0.3 \\ 0.3 & 0.0 & 0.0 & 0.7 \end{pmatrix}
$$

in which $T_1$ is for uniform noise with level 0.3, and $T_2$ is for pair flipping with noise level 0.3.

**Baselines.** We compare the proposed method with the following methods: (1) Standard, which trains the network in a standard manner, without any label resistance technique; (2) Forward correction (Patrini et al. 2017), which explicitly estimates the noise transition matrix to correct the training loss; (3) Decoupling (Malach & Shalev-Shwartz 2017), which trains two networks simultaneously and updates the parameters on selected data whose labels are possibly clean; (4) Coteaching (Han et al. 2018), which also trains two networks but exchanges their error information for network update; (5) MentorNet (Jiang et al. 2018), which learns a curriculum to filter out noisy data; (6) Forgetting (Arpit et al., 2017), which uses dropout to help deep models resist label noise.

**Experimental Setup.** For the classification of MNIST, CIFAR10 and CIFAR100, we use preactive resnet34 (He et al. 2016) as the backbone for all the methods. On ModelNet40, we use PointNet (Qi et al. 2017). We train the models for 180 epochs to ensure that all the methods have converged. We utilize RAdam (Liu et al. 2019) for the network optimization, and adopt batch size 128 for all the datasets. We use an initial learning rate of 0.001, which is decayed by 0.5 very 60 epochs. The experimental results are listed in Table 1. As is shown, our method outperforms the competing methods across the datasets under different noise settings.

Table 1: The classification accuracies of different methods.

| Data Set | Method | Noise Level of Uniform Flipping | | | | Noise Level of Pair Flipping | | |
|---|---|---|---|---|---|---|---|---|
| | | 0.2 | 0.4 | 0.6 | 0.8 | 0.2 | 0.3 | 0.4 |
| MNIST | Standard | $99.0 \pm 0.2$ | $98.7 \pm 0.4$ | $98.1 \pm 0.3$ | $91.3 \pm 0.9$ | $99.3 \pm 0.1$ | $99.2 \pm 0.1$ | $98.8 \pm 0.1$ |
| | Forget | $99.0 \pm 0.1$ | $98.8 \pm 0.1$ | $97.7 \pm 0.2$ | $62.6 \pm 8.9$ | $99.3 \pm 0.1$ | $96.5 \pm 2.0$ | $89.7 \pm 1.9$ |
| | Forward | $99.1 \pm 0.1$ | $98.7 \pm 0.2$ | $98.0 \pm 0.4$ | $89.6 \pm 4.8$ | $99.4 \pm 0.0$ | $99.2 \pm 0.2$ | $96.5 \pm 4.4$ |
| | Decouple | $99.3 \pm 0.1$ | $99.0 \pm 0.1$ | $98.5 \pm 0.2$ | $94.6 \pm 0.2$ | $99.4 \pm 0.0$ | $99.3 \pm 0.1$ | $99.1 \pm 0.2$ |
| | MentorNet | $99.2 \pm 0.2$ | $98.7 \pm 0.1$ | $98.1 \pm 0.4$ | $87.5 \pm 5.2$ | $98.6 \pm 0.4$ | $99.1 \pm 0.1$ | $98.9 \pm 0.1$ |
| | Coteach | $99.1 \pm 0.2$ | $98.7 \pm 0.3$ | $98.2 \pm 0.3$ | $95.7 \pm 0.7$ | $99.1 \pm 0.1$ | $99.0 \pm 0.2$ | $98.9 \pm 0.2$ |
| | **AdaCorr** | $\textbf{99.5} \pm \textbf{0.0}$ | $\textbf{99.4} \pm \textbf{0.0}$ | $\textbf{99.1} \pm \textbf{0.0}$ | $\textbf{97.7} \pm \textbf{0.2}$ | $\textbf{99.5} \pm \textbf{0.0}$ | $\textbf{99.6} \pm \textbf{0.0}$ | $\textbf{99.4} \pm \textbf{0.0}$ |
| CIFAR10 | Standard | $87.5 \pm 0.2$ | $83.1 \pm 0.4$ | $76.4 \pm 0.4$ | $47.6 \pm 2.0$ | $88.8 \pm 0.2$ | $88.4 \pm 0.3$ | $84.5 \pm 0.3$ |
| | Forget | $87.1 \pm 0.2$ | $83.4 \pm 0.2$ | $76.5 \pm 0.7$ | $33.0 \pm 1.6$ | $89.6 \pm 0.1$ | $83.7 \pm 0.1$ | $86.4 \pm 0.5$ |
| | Forward | $87.4 \pm 0.8$ | $83.1 \pm 0.8$ | $74.7 \pm 1.7$ | $38.3 \pm 3.0$ | $89.0 \pm 0.5$ | $87.4 \pm 1.1$ | $84.7 \pm 0.5$ |
| | Decouple | $87.6 \pm 0.4$ | $84.2 \pm 0.5$ | $77.6 \pm 0.1$ | $48.5 \pm 0.9$ | $90.6 \pm 0.3$ | $89.1 \pm 0.3$ | $86.3 \pm 0.5$ |
| | MentorNet | $90.3 \pm 0.3$ | $83.2 \pm 0.5$ | $75.5 \pm 0.7$ | $34.1 \pm 2.5$ | $90.4 \pm 0.2$ | $88.9 \pm 0.1$ | $83.3 \pm 1.0$ |
| | Coteach | $90.1 \pm 0.4$ | $87.3 \pm 0.5$ | $80.9 \pm 0.5$ | $25.0 \pm 3.6$ | $91.8 \pm 0.1$ | $89.9 \pm 0.2$ | $80.1 \pm 0.7$ |
| | **AdaCorr** | $\textbf{91.0} \pm \textbf{0.3}$ | $\textbf{88.7} \pm \textbf{0.5}$ | $\textbf{81.2} \pm \textbf{0.4}$ | $\textbf{49.2} \pm \textbf{2.4}$ | $\textbf{92.2} \pm \textbf{0.1}$ | $\textbf{91.3} \pm \textbf{0.3}$ | $\textbf{89.2} \pm \textbf{0.4}$ |
| CIFAR100 | Standard | $58.9 \pm 0.8$ | $52.1 \pm 1.0$ | $42.1 \pm 0.7$ | $20.8 \pm 1.0$ | $59.5 \pm 0.4$ | $52.9 \pm 0.6$ | $44.7 \pm 1.3$ |
| | Forget | $59.3 \pm 0.8$ | $53.0 \pm 0.2$ | $40.9 \pm 0.5$ | $7.7 \pm 1.1$ | $61.4 \pm 0.9$ | $54.6 \pm 0.6$ | $37.7 \pm 4.6$ |
| | Forward | $58.4 \pm 0.5$ | $52.2 \pm 0.3$ | $41.1 \pm 0.5$ | $20.6 \pm 0.6$ | $58.3 \pm 0.7$ | $53.2 \pm 0.6$ | $44.4 \pm 2.8$ |
| | Decouple | $59.0 \pm 0.7$ | $52.2 \pm 0.7$ | $40.2 \pm 0.4$ | $18.5 \pm 0.8$ | $60.8 \pm 0.7$ | $56.1 \pm 0.7$ | $48.4 \pm 1.0$ |
| | MentorNet | $63.6 \pm 0.5$ | $51.4 \pm 1.4$ | $38.7 \pm 0.8$ | $17.4 \pm 0.9$ | $64.7 \pm 0.2$ | $57.4 \pm 0.8$ | $47.4 \pm 1.7$ |
| | Coteach | $66.1 \pm 0.5$ | $60.0 \pm 0.6$ | $\textbf{48.3} \pm \textbf{0.1}$ | $16.1 \pm 1.1$ | $63.4 \pm 0.9$ | $57.6 \pm 0.3$ | $49.2 \pm 0.3$ |
| | **AdaCorr** | $\textbf{67.8} \pm \textbf{0.1}$ | $\textbf{60.2} \pm \textbf{0.8}$ | $46.5 \pm 1.2$ | $\textbf{24.6} \pm \textbf{1.1}$ | $\textbf{68.3} \pm \textbf{0.2}$ | $\textbf{61.1} \pm \textbf{0.5}$ | $\textbf{49.8} \pm \textbf{0.7}$ |
| ModelNet40 | Standard | $79.1 \pm 2.6$ | $75.3 \pm 3.3$ | $70.0 \pm 3.0$ | $57.9 \pm 2.3$ | $84.4 \pm 1.2$ | $82.3 \pm 1.3$ | $78.9 \pm 0.7$ |
| | Forget | $80.1 \pm 1.8$ | $73.9 \pm 0.6$ | $69.0 \pm 0.7$ | $26.2 \pm 4.8$ | $83.3 \pm 1.1$ | $62.0 \pm 3.0$ | $59.5 \pm 2.9$ |
| | Forward | $52.3 \pm 5.1$ | $49.4 \pm 6.8$ | $43.5 \pm 5.2$ | $28.2 \pm 5.5$ | $48.1 \pm 6.8$ | $48.0 \pm 3.7$ | $49.1 \pm 4.4$ |
| | Decouple | $82.5 \pm 2.2$ | $80.7 \pm 0.7$ | $72.9 \pm 1.0$ | $55.4 \pm 2.7$ | $85.7 \pm 1.4$ | $84.3 \pm 1.0$ | $80.5 \pm 2.4$ |
| | MentorNet | $86.5 \pm 0.5$ | $75.4 \pm 1.8$ | $70.9 \pm 1.9$ | $52.7 \pm 3.1$ | $83.7 \pm 1.8$ | $81.0 \pm 1.5$ | $79.3 \pm 2.1$ |
| | Coteach | $85.6 \pm 0.9$ | $84.2 \pm 0.8$ | $\textbf{81.8} \pm \textbf{1.1}$ | $68.9 \pm 2.8$ | $85.7 \pm 0.8$ | $79.1 \pm 3.0$ | $69.1 \pm 2.4$ |
| | **AdaCorr** | $\textbf{86.9} \pm \textbf{0.3}$ | $\textbf{85.1} \pm \textbf{0.6}$ | $78.6 \pm 1.4$ | $\textbf{72.1} \pm \textbf{1.1}$ | $\textbf{87.6} \pm \textbf{0.4}$ | $\textbf{84.6} \pm \textbf{0.5}$ | $\textbf{83.7} \pm \textbf{0.5}$ |

**Clothing 1M.** We also evaluate our method on a large scale Clothing 1M dataset (Xiao et al., 2015). We use pretrained resnet-50 and trained the model using SGD for 20 epochs. Our method has 71.47 accuracy. It outperforms Standard (68.94) and Forward (69.84). Other baselines (Forget, Decouple, MentorNet, Coteach) did not report result on this dataset.

## 4 ABLATION STUDY AND DISCUSSION

We conduct ablation study to see the significance of our contributions. We compare our method (LRT + $L_{ce}$+$L_{retro}$) with two baselines: our method without the retroactive loss (LRT+$L_{ce}$) and using cross-entropy loss only without LRT-Correction ($L_{ce}$ Only). We report the test accuracy on CIFAR10 in Table 2. We observe that adding LRT Label Correction to $L_{ce}$ alone helps improve the performance significantly. The numbers inside the parenthesis are the percentage of correct labels after the training and label correction. We observe that the LRT Correction corrected a large portion of noise labels. Also we observe adding the retroactive loss improves the method further in both test accuracy and label correction rates.

Table 2: Effect of LRT Correction and $L_{retro}$. The experiments are performed on CIFAR10 (accuracy in %). The number in parenthesis denotes the rate of correct labels after training.

| Method | uniform 0.2 | uniform 0.4 | uniform 0.6 | uniform 0.8 | pair 0.4 |
|---|---|---|---|---|---|
| $L_{ce}$ Only | 87.5(80.0) | 83.1(60.0) | 76.4(40.0) | 47.6(20.0) | 84.5(60.0) |
| LRT+$L_{ce}$ | 91.1(95.7) | 87.9(91.2) | 80.7(81.7) | 47.3(47.1) | 87.5(91.9) |
| LRT+$L_{ce}$+$L_{retro}$ | 91.0(95.7) | 88.7(90.5) | 81.2(82.5) | 49.2(49.0) | 89.2(91.3) |

We evaluate how different hyperparameters affect the performance of our method. We compare our method with different $m$, the length of the burn-in stage. We start introducing the retroactive loss after $m$ epochs, and start label corrections after $m + 10$ epochs. The final testing accuracies are shown in Table 3. We observe the performance of our method is rather robust w.r.t. different $m$'s. We choose $m = 20$ in this data set (CIFAR10) and similarly in other datasets.

Table 3: Effect of different $m$'s. The experiments are performed on CIFAR10 (accuracy in %). The number in parenthesis denotes the rate of correct labels after training.

| Noisy Type | Epoch 15 | Epoch 20 | Epoch 25 | Epoch 30 | Epoch 35 | Epoch 40 |
|---|---|---|---|---|---|---|
| uniform 0.4 | 87.6(90.1) | 88.7(90.5) | 87.4(90.7) | 86.7(90.6) | 84.8(88.7) | 84.1(87.2) |
| uniform 0.6 | 79.4(81.0) | 81.2(82.5) | 80.9(81.9) | 79.3(81.9) | 79.1(81.8) | 78.1(81.7) |
| pair 0.4 | 75.8(80.0) | 89.2(80.1) | 90.8(87.0) | 89.2(89.3) | 88.2(90.1) | 86.7(90.1) |

We also evaluate the performance on different $\Delta$'s. $\Delta$ is the unknown value for our likelihood ratio test. It controls how aggressive we would like to correct the labels. From Table 4, we observe a bigger $\Delta$ tends to give better results (as it is less aggressive in correcting labels). We observe 1/1.2 is the best one for CIFAR10. Similar values of optimal $\Delta$ are found in other data sets.

Table 4: Effect of $\Delta$. The experiments are performed on CIFAR10, and the number in parenthesis denotes the rate of correct labels after flipping.

| Noisy Type | 1/1.0 | 1/1.2 | 1/1.5 | 1/2.0 | 1/2.5 | 1/3.0 |
|---|---|---|---|---|---|---|
| uniform 0.4 | 88.3(91.7) | 88.7(90.5) | 83.0(89.3) | 77.1(86.7) | 75.2(85.1) | 75.5(84.0) |
| uniform 0.6 | 79.9(81.0) | 81.2(82.5) | 80.9(81.9) | 79.3(81.9) | 79.1(81.8) | 78.1(81.7) |
| pair 0.4 | 88.2(92.2) | 89.2(80.1) | 84.4(89.8) | 77.0(86.8) | 76.4(85.1) | 77.0(84.1) |

In general, we observe our hyperparameters are rather consistent across different datasets. This reveals a better generalization power of our method over other datasets and noise patterns. We believe this is due to the principled approach we take in label cleaning.

In Table 5 we show the performance of different methods trained on clean dataset. We observe that when without label noise, all the methods achieve very similar performance. However, note that on CIFAR10, CIFAR100 and ModelNet40, the performances of MentorNet and Forward are slightly inferior to the standard approach. The reason could be that MentorNet is designed specifically for handling noisy data, and would behave too conservatively under clean data setting; for Forward, although the data is clean, the estimated noise transition matrix is imperfect and still indicates a certain level of noise, leading to inferior performance. AdaCorr remains competitive in these experiments.

Table 5: The performance of different methods trained on clean dataset.

| Dataset | Standard | Forget | Forward | Decouple | MentorNet | Coteach | AdaCorr |
|---|---|---|---|---|---|---|---|
| MNIST | $99.5 \pm 0.1$ | $99.5 \pm 0.1$ | $99.5 \pm 0.1$ | $99.5 \pm 0.0$ | $99.4 \pm 0.2$ | $99.5 \pm 0.0$ | $99.4 \pm 0.3$ |
| CIFAR10 | $93.6 \pm 0.3$ | $93.5 \pm 0.3$ | $91.2 \pm 0.4$ | $93.7 \pm 0.4$ | $91.5 \pm 0.4$ | $93.6 \pm 0.2$ | $93.9 \pm 0.2$ |
| CIFAR100 | $72.0 \pm 0.4$ | $72.0 \pm 0.3$ | $70.1 \pm 0.3$ | $72.0 \pm 0.3$ | $71.4 \pm 0.3$ | $72.2 \pm 0.5$ | $74.1 \pm 0.0$ |
| ModelNet40 | $87.6 \pm 0.5$ | $87.7 \pm 0.5$ | $60.3 \pm 0.6$ | $88.4 \pm 0.6$ | $85.0 \pm 0.5$ | $85.9 \pm 0.4$ | $88.4 \pm 0.3$ |

In Figure 3, we draw convergence curves on CIFAR10 with Uniform 0.4 noise level. On the left, we show the curves of our proposed AdaCorr method. The model continues to fit with the corrected labels ($y_{new}$ and the test accuracy on clean labels does not drop. This shows that the model and the label correction are improving in a harmonic fashion and do not collapse. On the right, we show the curves of the Standard model. Without label correction, the model overfits with the noisy label and loses on test data catastrophically.

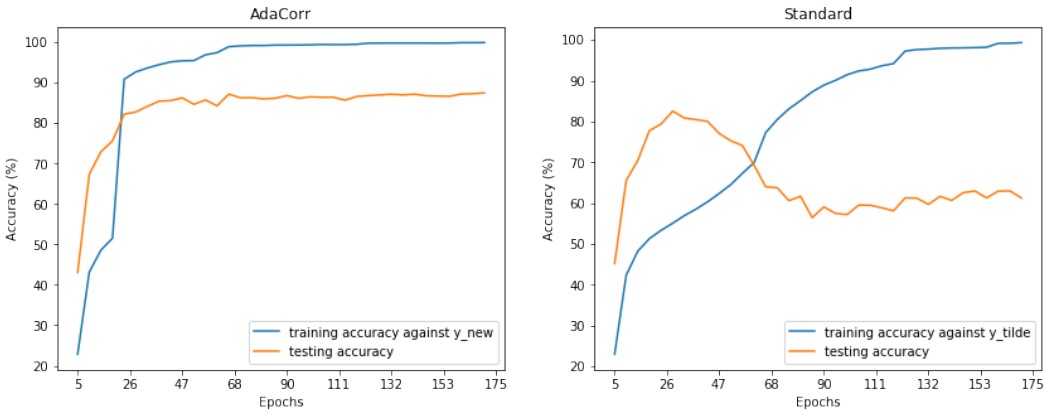

Figure 3: Convergence curves for CIFAR10 with uniform 0.4 noisy level. Left: AdaCorr - training accuracy evaluated against the corrected label ($y_{new}$), testing accuracy against clean label. Right: Standard - training accuracy against noisy label ($\widetilde{y}$) and testing accuracy against clean label.

## 5 CONCLUSION

We propose a label correction algorithm to combat label noise. The correction algorithm performs a likelihood ratio test for each input label to decide whether to correct the label. Theoretically, we prove that our method corrects noisy labels with high probability. By combining with deep neural network training with a novel retroactive loss, our method can produce models robust to different noise patterns. Experiments on various datasets show that our method outperforms state-of-the-arts and is robust to hyperparameters.

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

## A  PROOF OF THEOREM 1

We restate the theorem below for completeness.

**Theorem 1** Assume $\eta$ satisfies the Tsybakov condition with constants $C > 0$ and $\lambda > 0$. Let $h^*$ denote the Bayes optimal classifier, and $\widetilde{\eta}$ the noisy conditional probability $\widetilde{\eta}(\boldsymbol{x}) = (1 - \tau_{01} - \tau_{10})\eta(\boldsymbol{x}) + \tau_{01}$. Let $f$ be any classifier such that $||f - \widetilde{\eta}||_\infty \leq \epsilon$ for some $\epsilon > 0$. Let $\Delta = \min(\frac{1+\tau_{01}-\tau_{10}}{1+\tau_{10}-\tau_{01}}, \frac{1+\tau_{10}-\tau_{01}}{1+\tau_{01}-\tau_{10}})$, and let $\Delta' > 0$ be a constant such that $\Delta' \in [\Delta - \epsilon, \Delta + \epsilon]$. If $\widetilde{y}_{new}$ denotes the output of the `LRT-Correction` with $\widetilde{y}$, $\boldsymbol{x}$, $f$, and the given $\Delta'$, then

$$\Pr_{(x,y)\sim D}(\widetilde{y}_{new} \neq h^*) \leq C \left( \left| \frac{\tau_{10} - \tau_{01}}{2(1 - \tau_{10} - \tau_{01})} \right| + 2\epsilon \right)^\lambda.$$

If $\tau_{01} = \tau_{10}$, then $\Pr_{(x,y)\sim D}(\widetilde{y}_{new} \neq h^*) = C (2\epsilon)^\lambda$.

To prove this theorem, we first prove two lemmas. In the following we will state and prove the lemmas, and then continue to finish the proof of Theorem 1.

**Lemma 2.** *Assume $\eta$ satisfies the Tsybakov condition with constants $C > 0$ and $\lambda > 0$. Let $h^*$ denote the Bayes classifier. If $f$ depends linearly on $\eta$, i.e., $f(\boldsymbol{x}) = a\eta(\boldsymbol{x}) + b$, $a, b > 0$, and $a, b$ and $u$ are known, and $\Delta$ is chosen as in Table 6, then*

- *If `LRT-Correction`($f$,($\boldsymbol{x}, \widetilde{y}$), $a$, $b$,) flips the label of $\widetilde{y}$, then $\widetilde{y}_{new} = h^*(\boldsymbol{x})$, and*
- $\Pr_{(\boldsymbol{x},y)\sim D}(\widetilde{y}_{new}(x, \widetilde{y}) \neq h^*(\boldsymbol{x})) = 0.$

**Lemma 3.** *Assume $\eta$ satisfies the Tsybakov condition with constants $C > 0$ and $\lambda > 0$. Let $h^*$ denote the Bayes classifier. Let $\widetilde{\eta}(\boldsymbol{x}) = (1 - \tau_{01} - \tau_{10})\eta(\boldsymbol{x}) + \tau_{01}$ denote the corrupted Bayesian classifier. If $\Delta = \min(\frac{1+\tau_{01}-\tau_{10}}{1+\tau_{10}-\tau_{01}}, \frac{1+\tau_{10}-\tau_{01}}{1+\tau_{01}-\tau_{10}})$, then*

Table 6: Values of $\Delta$.

| $\widetilde{y}$ | $f < b + a/2$ | $f > b + a/2$ |
|---|---|---|
| $\widetilde{y} = 1$ | $\frac{a+2b}{2-a-2b}$ | $0$ |
| $\widetilde{y} = 0$ | $0$ | $\frac{2-a-2b}{a+2b}$ |

- *If* `LRT-Correction`$(f,(\boldsymbol{x},\widetilde{y}), \Delta)$ *flips the label of $\widetilde{y}$, then $\widetilde{y}_{new} = h^*(\boldsymbol{x})$,*
- $\Pr_{(\boldsymbol{x},y)\sim D}(\widetilde{y}_{new}(x,\widetilde{y}) \neq h^*(\boldsymbol{x})) \leq C\left(\left|\frac{\tau_{01}-\tau_{10}}{2(1-\tau_{10}-\tau{10})}\right|\right)^\lambda$, *and*
- *If $\tau_{01} = \tau_{10}$, $\Pr_{(\boldsymbol{x},y)\sim D}(\widetilde{y}_{new}(x,\widetilde{y}) \neq h^*(\boldsymbol{x})) = 0$.*

**Proof of Lemma 2:**

We begin with some observations:

**Observation 1:** If $\widetilde{y} = \mathbb{1}(f > b + a/2)$, the algorithm never flips. This is because $\Delta = 0$ in all such cases, and since the likelihood ratios $f/(1 - f)$ or $(1 - f)/f$ are always positive, $\widetilde{y}$ is not flipped.

**Observation 2:** $f(\boldsymbol{x}) < b + a/2$ if, and only if, $\eta(\boldsymbol{x}) < 1/2$. This follows by a straightforward calculation using $f(\boldsymbol{x}) = a\eta(\boldsymbol{x}) + b$.

*In the remainder of this proof, we assume that $\tau_{10} > \tau_{01}$. The case $\tau_{10} < \tau_{01}$ follows by symmetry.*

**Observation 3:** If $\widetilde{y}(\boldsymbol{x}) = 1$ and $f(\boldsymbol{x}) < b + a/2$ (so a non-trivial test (unlike ones in Observation 1) is performed), then $f(\boldsymbol{x})/(1 - f(\boldsymbol{x})) < \Delta(\widetilde{y} = 1, f(\boldsymbol{x}) < b + a/2)$ if, and only if, $\eta(\boldsymbol{x}) < 1/2$. Similarly if $\widetilde{y}(\boldsymbol{x}) = 0$ and $f(\boldsymbol{x}) > b + a/2$ then $(1 - f(\boldsymbol{x}))/f(\boldsymbol{x}) < \Delta(\widetilde{y} = 0, f(\boldsymbol{x}) > b + a/2)$ if, and only if, $\eta(\boldsymbol{x}) > 1/2$.

**Proof of Observation 3:** First notice from the table that $\Delta(\widetilde{y} = 1, f(\boldsymbol{x}) < 1/2)$ is the reciprocal of $\Delta(\widetilde{y} = 0, f(\boldsymbol{x}) > 1/2)$. Since the likelihood test ratio is also the reciprocal in the two situations, proving one statement suffices. We prove the first statement. For convenience we denote $\Delta(\widetilde{y} = 1, f(\boldsymbol{x}) < 1/2)$ by $\Delta$.

$$\frac{f(\boldsymbol{x})}{1 - f(\boldsymbol{x})} < \Delta \tag{4}$$

$$\iff \quad f(\boldsymbol{x}) < \Delta/(1 + \Delta) \tag{5}$$

$$\iff \quad f(\boldsymbol{x}) < \frac{\frac{a+2b}{2-a-2b}}{1 + \frac{a+2b}{2-a-2b}} \tag{6}$$

$$\iff \quad f(\boldsymbol{x}) < \frac{a + 2b}{2} \tag{7}$$

$$\iff \quad \eta(\boldsymbol{x}) < \frac{\frac{a+2b}{2} - b}{a} \tag{8}$$

$$\iff \quad \eta(\boldsymbol{x}) < 1/2. \tag{9}$$

We can now prove the first assertion of Lemma 2 (under the assumption that $\tau_{10} > \tau_{01}$). That is, if *LRT-Correction(f,($\boldsymbol{x},\widetilde{y}$), a, b)* flips the label of $\widetilde{y}$, then $\widetilde{y}_{new} = h^*(\boldsymbol{x})$. Let us first consider the case $\widetilde{y}(\boldsymbol{x}) = 1$. If $f(\boldsymbol{x}) > b + a/2$, then it implies that $\eta(\boldsymbol{x}) > 1/2$ (Observation 2). The corresponding $\Delta$ in the table is 0, meaning that no test is performed, and $\widetilde{y}_{new}(\boldsymbol{x},\widetilde{y}) = \widetilde{y}(\boldsymbol{x}) = h^*(\boldsymbol{x})$. On the other hand, if $f(\boldsymbol{x}) < b + a/2$, we perform a likelihood ratio test on $f/(1 - f)$, which is less than the corresponding value of $\Delta$ iff $\eta(\boldsymbol{x}) < 1/2$ (Observation 3), so again $\widetilde{y}_{new}(\boldsymbol{x},\widetilde{y}) = \widetilde{y}(\boldsymbol{x}) = h^*(\boldsymbol{x})$. The case $\widetilde{y}(\boldsymbol{x}) = 0$ can be analyzed analogously.

Thus the only points $\boldsymbol{x}$ where we may not match $h^*(\boldsymbol{x})$ are ones where the flipping algorithms does not flip, and the result mismatches with $h^*$. By the above observations, the following is where this happens.

**Observation 4:** Assume $\tau_{10} > \tau_{01}$. Then

$$\{\boldsymbol{x} : \widetilde{y}_{new}(\boldsymbol{x},\widetilde{y}) \neq h^*(\boldsymbol{x})\} = \{\boldsymbol{x} : \widetilde{y}(\boldsymbol{x}) = 0, f(\boldsymbol{x}) < b + a/2, \eta(\boldsymbol{x}) > 1/2\}.$$

However, according to Observation 2, $f(\boldsymbol{x}) < b + a/2$ implies $\eta(\boldsymbol{x}) < 1/2$. Thus this case cannot happen. Hence $\Pr_{(\boldsymbol{x},y)\sim D}(\widetilde{y}_{new}(x,\widetilde{y}) \neq h^*(\boldsymbol{x})) = 0$, and Lemma 2 is proved.

**Proof of Lemma 3**

This proof is an extension of the proof of Lemma 2. All the steps are analogous, except in Observation 4, we get the following

**Observation 5:** Assume $\tau_{10} > \tau_{01}$. Then

$$\{\boldsymbol{x} : \widetilde{y}_{new}(\boldsymbol{x},\widetilde{y}) \neq h^*(\boldsymbol{x})\} = \{\boldsymbol{x} : \widetilde{y}(\boldsymbol{x}) = 0, \widetilde{\eta}(\boldsymbol{x}) < 1/2, \eta(\boldsymbol{x}) > 1/2\}.$$

This measure is non zero. However, it is bounded above by the measure of the set $\{\widetilde{\eta}(\boldsymbol{x}) < 1/2, \eta(\boldsymbol{x}) > 1/2\}$. Since $\widetilde{\eta}(\boldsymbol{x}) = (1 - \tau_{01} - \tau_{10})\eta(\boldsymbol{x}) + \tau_{01}$, this is the same as the measure of the set $\{\boldsymbol{x} : 1/2 < \eta(\boldsymbol{x}) < \frac{1/2 - \tau_{01}}{1 - \tau_{10} - \tau_{01}}$. By the Tsybakov condition, this measure is at most $\left(\frac{\tau_{10} - \tau_{01}}{2(1 - \tau_{10} - \tau_{01})}\right)^{\lambda}$. This was for the case $\tau_{10} > \tau_{01}$, and when $\tau_{10} < \tau_{01}$ we can check by analogous calculation that the numerator is reversed. Hence we have proved

$$\Pr_{(\boldsymbol{x},y)\sim D}(\widetilde{y}_{new}(x,\widetilde{y}) \neq h^*(\boldsymbol{x})) \leq C \left(|\frac{\tau_{01} - \tau_{10}}{2(1 - \tau_{10} - \tau 10)}|\right)^{\lambda}.$$

The last assertion of Lemma 3 follows by just substituting $\tau_{01} = \tau_{10}$ in the above formula. Hence Lemma 3 is proved.

**Proof of Theorem 1**

Let $A$ denote the event $\left(\frac{f}{1-f} < \Delta'\right)$, and $B$ denote the event $\left(\frac{1-f}{f} < \Delta'\right)$. In the following we describe the cases when our algorithm results (flips or does not flip) in a label inconsistent with $h^*$.

- $(\widetilde{y} = 1, f < 1/2, \eta > 1/2, \text{LRT flips})$. According to the LRT scheme, this happens with probability at most $\Pr(A \cap (\eta > 1/2))$.
- $(\widetilde{y} = 1, f < 1/2, \eta < 1/2, \text{LRT does not flip})$. According to the LRT scheme, this happens with probability at most $\Pr(A^c \cap (\eta < 1/2))$. Here $A^c$ denotes the complement of the event $A$.
- $(\widetilde{y} = 1, f > 1/2, \eta > 1/2, \text{LRT flips})$. According to the LRT scheme, this happens with probability at most $\Pr(A \cap (\eta > 1/2))$.
- $(\widetilde{y} = 1, f > 1/2, \eta < 1/2, \text{LRT does not flip})$. According to the LRT scheme, this happens with probability at most $\Pr(A^c \cap (\eta < 1/2))$.
- $(\widetilde{y} = 0, f < 1/2, \eta > 1/2, \text{LRT does not flip})$. According to the LRT scheme, this happens with probability at most $\Pr(B^c \cap (\eta > 1/2))$.
- $(\widetilde{y} = 0, f < 1/2, \eta < 1/2, \text{LRT flips})$. According to the LRT scheme, this happens with probability at most $\Pr(B \cap (\eta < 1/2))$.
- $(\widetilde{y} = 0, f > 1/2, \eta < 1/2, \text{LRT flips})$. According to the LRT scheme, this happens with probability at most $\Pr(B \cap (\eta < 1/2))$.
- $(\widetilde{y} = 0, f > 1/2, \eta > 1/2, \text{LRT does not flip})$. According to the LRT scheme, this happens with probability at most $\Pr(B^c \cap (\eta > 1/2))$.

In all of these cases, one observes (after using the fact that $||f - \widetilde{\eta}||_{\infty} \leq \epsilon$ that the probability of these 8 events is either of the form

- $\Pr\left(1/2 < \eta(\boldsymbol{x}) < \frac{\frac{1}{1+\Delta'} - \tau_{01}}{1 - \tau_{01} - \tau_{10}} + \epsilon\right)$, or

- $\Pr\left(\frac{\frac{1}{1+\Delta'} - \tau_{01}}{1 - \tau_{01} - \tau_{10}} - \epsilon < \eta < 1/2\right)$, or

- $\Pr\left(1/2 < \eta(\boldsymbol{x}) < \frac{\frac{\Delta'}{1+\Delta'} - \tau_{01}}{1 - \tau_{01} - \tau_{10}} + \epsilon\right)$, or

- $\Pr\left(\frac{\frac{\Delta'}{1+\Delta'} - \tau_{01}}{1 - \tau_{01} - \tau_{10}} - \epsilon < \eta < 1/2\right)$.

Next, we observe that $|\frac{1}{1+\Delta'} - \frac{1}{1+\Delta}| = |\frac{\Delta - \Delta'}{(1+\Delta')(1+\Delta)}| \leq \epsilon$. Also, $|\frac{\Delta'}{1+\Delta'} - \frac{\Delta}{1+\Delta}| \leq \epsilon$.

Using these observations, the four expressions can be bounded by the analogous four expressions found by replacing $\Delta'$ by $\Delta$.

- $\Pr\left(1/2 < \eta(\boldsymbol{x}) < \frac{\frac{1}{1+\Delta} - \tau_{01}}{1 - \tau_{01} - \tau_{10}} + 2\epsilon\right)$, or
- $\Pr\left(\frac{\frac{1}{1+\Delta'} - \tau_{01}}{1 - \tau_{01} - \tau_{10}} - 2\epsilon < \eta < 1/2\right)$, or
- $\Pr\left(1/2 < \eta(\boldsymbol{x}) < \frac{\frac{\Delta'}{1+\Delta'} - \tau_{01}}{1 - \tau_{01} - \tau_{10}} + 2\epsilon\right)$, or
- $\Pr\left(\frac{\frac{\Delta'}{1+\Delta'} - \tau_{01}}{1 - \tau_{01} - \tau_{10}} - 2\epsilon < \eta < 1/2\right)$.

Assuming Tsybakov condition, one can bound each of these probabilities by $C\left(\left|\frac{\tau_{10} - \tau_{01}}{2(1 - \tau_{10} - \tau_{01})}\right| + 2\epsilon\right)^\lambda$. Notice that although there are $8$ cases, they are mutually disjoint, so the final failure probability is bounded by the maximum of these $8$ cases, also of the same form. Hence we have proved

$$\Pr_{(x,y)\sim D}(\widetilde{y}_{new} \neq h^*) \leq C\left(\left|\frac{\tau_{10} - \tau_{01}}{2(1 - \tau_{10} - \tau_{01})}\right| + 2\epsilon\right)^\lambda.$$

which finishes the proof of Theorem 1.

