# OpenReview forum: "Label Cleaning with Likelihood Ratio Test"
_ICLR.cc/2020/Conference — Reject_

### Official Review · AnonReviewer3 · 2019-10-21
**Official Blind Review #3**

**Rating:** 8

**Review:**

Label noise widely exists in the large-scale data sets. This paper proposes a novel approach that directly cleans labels in order to train a high quality model. The proposed method leverages statistical principles to correct data labels and has a theoretical guarantee of the correctness. In particular, a likelihood ratio test (LRT) to flip the labels of training data is used, and it can prove that the LRT label correction algorithm is guaranteed to flip the label so it is consistent with the true Bayesian optimal classifier with high probability. The experimental results on several benchmark data sets show that the proposed method is promising. Overall, this paper could be a significant algorithmic contribution. But I also have some minor concerns:
[1] The theoretical analysis and the experimental results are both well organized in the paper. How about the time complexity of the proposed method. If the authors can show the time cost in the paper, I will much more agree with the paper.
[2] In the experimental parts, the convergence curve of the proposed method during the training epochs may be better to prove the theoretical analysis.
[3]The details of the compared methods should be given, and it will be better to give the results without any noise labels. In this way, the confidence of the paper will be further improved.


**Experience Assessment:**

I have read many papers in this area.

**Review Assessment: Checking Correctness Of Derivations And Theory:**

I assessed the sensibility of the derivations and theory.

**Review Assessment: Checking Correctness Of Experiments:**

I did not assess the experiments.

**Review Assessment: Thoroughness In Paper Reading:**

I read the paper at least twice and used my best judgement in assessing the paper.

---

> ### Author Response · Authors · 2019-11-15
> **Reply to Reviewer #3**
>
> Thank you for reviewing our paper and providing constructive feedback. We address your concerns below.
>
> ** [1]. Time complexity.
> A: For each data at each iteration, despite the number of classes, the flipping checks two posterior probability (of the noisy label \tilde{y} and the model prediction label y*, see Procedure 2 of the revised version for details). The complexity is O(1). In practice, the time is still dominated by forwarding and backpropagation. The retroactive loss is just like another cross-entropy loss, except for it is comparing with the model prediction at a previous epoch.
>
> In experiments, our method takes less than twice longer than a base model (Standard) per iteration. On CIFAR10, to process each mini-batch (size 128),our algorithm takes 0.4 seconds while Standard takes 0.25 seconds. On Clothing 1M,  for each  mini-batch (size 32), our algorithm takes  0.33  seconds  whereas Standard takes 0.29 seconds. We use nvdia GTX 1080TI GPUs. A similar discussion has been provided in replying to Reviewer #2's comments.
>
> ** [2]. Provide convergence curve.
> A: excellent suggestion! The convergence curves of our method and the baseline Standard have been provided and discussed in Figure 3 of the revised paper.
>
> ** [3]. Results on data without noise.
> A: thanks for the suggestion. We will provide more details of the baselines in the final version. We evaluated all models on a clean dataset and reported results in Table 5 of Section 4.

---

### Official Review · AnonReviewer1 · 2019-10-23
**Official Blind Review #1**

**Rating:** 3

**Review:**

This paper proposes a new method for correcting training label noise, using a likelihood ratio test on the predicted probability of the classifier trained on the noisy labels, and then proposes an algorithm for iteratively cleaning training labels and retraining a model.  The paper also provides a theoretical guarantee for the probability of correctly re-labeling the training set, and provides empirical results showing that the proposed method significantly outperforms existing approaches for handling noisy training labels.

Overall, I think the empirical results appear very strong, but think this paper is below the acceptance threshold due to three factors, in ranked order:
- (1) The theoretical guarantee, which is positioned as a core contribution of the paper (and in fact claims it as "the first to correct labels with theoretical guarantees", which is not true), is based on assumptions that seem overly strong; these are somewhat relaxed in a "Remark", but this seems unproven and is a confusing presentation regardless.
- (2) The theoretical bound itself is somewhat vacuous as it contains several constant factors that seem very material to the bound, but totally opaque to the reader.
- (3) The experiments are very strong overall, which is a major plus for the paper; however, there are some questions about the hyperparameter tuning and some other points where more clarity could improve the strength of the empirical results

Regarding (1):
As a reader, my first natural reaction was to worry about circularity/degeneracy in the proposed method: basically, we are using the confidence (as a ratio of predicted cond. probabilities) of the model trained on the corrupted labels to correct those labels... if the labels are so corrupted that the model is also way off, then intuitively, this method should not work.  I was wondering about how this situation would be bounded / handled.

It turns out in Thm 1 that an incredibly strong assumption is made, namely that the model trained on the corrupted labels, f, is a linear function of the true model, with constants a, b known to some small degree of error epsilon (note that the theorem statement says that these constants are unknown- but it then assumes that \Delta, which is set based on a and b, is known up to \epsilon error).  This seems like an incredibly strong assumption- and no context / motivation is given about why it should be taken as reasonable.

Then, immediately after the Theorem, "Remark 2" states that this condition is not actually needed at all- but (a) then why not just strike it from Thm 1 statement, and (b) there does not seem to be any proof of this Remark in the appendix (where the proof of the main theorem itself is closer to a sketch than a standard proof...).

Regarding (2):
The bound produced in Thm 1 seems somewhat vacuous: letting \tau_{01} = \tau_{10}, then the probability of the label correction being erroneous is bounded by 8C(O(\epsilon))^\lambda.  This quantity is presumably in (0,1/2], so it's a small range to start... but it seems hard to get anything from this bound without some idea of what the constant factors (C, and those hidden in O(\epsilon)) and \lambda are.  In particular, as presented, it seems implausible that \epsilon- the error in specifying the \Delta threshold- gets that small, in which case these constants become very important to know!  Another way of phrasing this remark: many theoretical bounds have lots of unknown constant factors, but are ultimately just trying to expose some scaling with respect to one parameter, e.g. number of data points, and therefore the constants don't need to be known that well for the statement to have some value.  This doesn't exactly seem to be the case here- therefore it seems hard to extract something from this statement (even ignoring the strong assumptions it is predicated on).

Regarding (3):
Overall, I think the empirical performance reported in Table 2, and overall thoroughness of the ablation in Section 4, are major strong points for the paper- the performance is very impressive!  However I have a few questions, clarification of which would be very helpful in my mind:
- (i) A major issue that seems to be raised in the earlier sections is that there is a hyperparameter \Delta--the threshold for the likelihood ratio test--that everything depends on, and must be chosen empirically.  Table 5 shows that the effect of choosing it is not crazy, but also clearly not insignificant.  My question is: how is it chosen?  On the validation dataset?  And is this validation dataset also corrupted in the same way as the training dataset?  If not, that seems like a major whole in the setup.
- (ii) I also have a high level question for understanding: how is it possible for the various approaches to do so well with 0.6 and 0.8 noise level of uniform flipping?  In the p=0.8 noise model, for example, the probability of a data point getting flipped to *any individual wrong label* is *greater than that of it being the correct label*.  How is it possible to learn a model based on such a dataset?  Was there some kind of pre-training?  Was the validation set not corrupted?  I don't conceptually understand how the results shown are possible...?

Overall, I think points in (1) and (3) could be helped with additional clarification and contextualization, and possibly (2) as well.

**Experience Assessment:**

I have published in this field for several years.

**Review Assessment: Checking Correctness Of Derivations And Theory:**

I assessed the sensibility of the derivations and theory.

**Review Assessment: Checking Correctness Of Experiments:**

I carefully checked the experiments.

**Review Assessment: Thoroughness In Paper Reading:**

I read the paper thoroughly.

---

> ### Author Response · Authors · 2019-11-15
> **Reply to Reviewer #1**
>
> Thank you for constructive comments. We address your concerns below. We split each of your comments into sub-comments such as (1-i) and (1-ii) for clarity.
>
> ** (1-i). "the first to correct labels with theoretical guarantees", is not true.
> A: We are not sure which previous work you are referring to. As we originally stated, we are the first to prove theoretical guarantees for *label-correction / data-selection* deep learning methods in noisy label problems. There might be theoretical guarantees for the noisy label problem, but not for the label-correction/data-selection type solutions.
>
> ** (1-ii). if the labels are so corrupted that the model is also way off, then intuitively, this method should not work.
> A: The trick is to use the model to correct labels way before it overfits the noise. We start flipping at m which is the phase transition time between fitting common patterns and memorizing noise (Arpit et al. ICML 2017). This is when the model is still trustable, although not completely fitting the training set. As labels get corrected, the model will stay in a good condition and will not collapse. See our discussion regarding comment 3 of Reviewer #2.
>
> ** (1-iii). In Thm 1, assumption on f to be linear to the true function is too strong. Why not strike it? Remark 2 is confusing.
> A: Agreed. Indeed we removed this linear assumption and Remark 2 completely. We updated Theorem 1 in the revised paper. We do not need to assume f is linear with respect to the Bayesian optimal. This assumption is just needed in one of the lemmas (Lemma 2) that studies this special case and is used to build up to the theorem. In the main theorem, all we require is that f be within \epsilon of the noisy classifier \tilde{\eta}, not the true one.
>
> ** (2). Hidden constant factors can be large.
> A: We completely understand your concern. We calculated the hidden constant in O(\epsilon). It is indeed just 2. We explicitly stated it in the revised Theorem 1. The calculation is included in the proof (Appendix A).
>
> As for the constants C and \lambda, the Tsybakov condition is a classic assumption about the data distribution. It has been well accepted and has been used in many machine learning papers to establish bounds, e.g., [1-5]. Many of the results are bounds on generalization error, which is in a very similar range as ours, (0,1/2].
>
> Intuitively, C and \lambda quantifies how well can an ideal classifier separate the data. \lambda=0 is the ``"flat"/worst case for a classifier. \lambda = \infty is the threshold/best case for a classifier. Also the smaller C is, the better the classifier is. As a sanity check, we can prove that for a mixture of two Gaussians in 1D domain (one Gaussian for each class), \lambda=1/2. C depends on the distance between the two Gaussians' centers and their width.
>
> ** (3-i). Choosing \Delta.
> A: Our theory provides guidance on choosing \Delta. We use a same setting of \Delta across all datasets and noise patterns. \Delta is in the range of [0,1]. We start from setting \Delta to be close to 1 (i.e., 1/1.2), and then gradually decrease it to 0 during the training. At a later stage of the training, using smaller \Delta means being more conservative in flipping labels. This provides a safety mechanism and ensures the choice of initial \Delta to be less sensitive. We had a similar discussion in replying to Reviewer #2's comment.
>
> ** (3-ii) How can the method work at uniform noise level 0.8?
> A: there seems to be a misunderstanding. Uniform noise level 0.8 means a label has 80% chance to be flipped to one of the other wrong labels. The 80% chance is evenly split between all the possible wrong labels. For example, for a 10-class CIFAR10 dataset, a data whose true class is 1 will have 80% / 9 = 8.9% to be flipped to class 2 (or class 3, 4, ..., 10). This is still smaller than the 20% chance of the data remaining with label 1. During flipping, we only compare two labels at each time. Thus, the true label class is still dominating and we still have a good chance to recover it. The theoretical limit of our method is uniform noise level 0.9. This is when a data has equal chance of 10% to be any label. The true label loses its dominance.
>
> ** References:
> [1]. Henry W. J. Reeve, Ata Kabán, Fast Rates for a kNN Classifier Robust to Unknown Asymmetric Label Noise, ICML, 2019
> [2]. Henry W. J. Reeve, Ata Kabán, Classification with unknown class conditional label noise on non-compact feature spaces. COLT, 2019
> [3]. Kamalika Chaudhuri, Sanjoy Dasgupta, Rates of Convergence for Nearest Neighbor Classification, NeurIPS, 2014
> [4]. Yichong Xu, Hongyang Zhang, Kyle Miller, Aarti Singh and Artur Dubrawski, Noise-Tolerant Interactive Learning Using Pairwise Comparisons, NeurIPS, 2017
> [5]. Yining Wang, Aarti Singh, Noise-Adaptive Margin-Based Active Learning and Lower Bounds under Tsybakov Noise Condition, AAAI, 2016

---

### Official Review · AnonReviewer2 · 2019-10-24
**Official Blind Review #2**

**Rating:** 3

**Review:**

This paper proposes a label correction approach based on a likelihood ratio test, for robust training of deep neural networks against label noise. First, this paper introduces the LRT-Correction procedure, which is the main component of the proposed label correction approach. LRT-Correction uses the current model prediction to run a likelihood ratio test and flip labels when they are rejected. The decision is made by comparing the likelihood test results with a predefined value \Delta. Then they introduce the full algorithm, AdaCorr, where the LRT-Correction procedure serves as an inner loop for the label correction. Lastly, there are experiments done on four datasets to conclude the superior performance of the proposed AdaCorr in contrast to several existing methods.

Overall, this paper proposes a new label correction approach based on a likelihood ratio test. Standard experiments show that the proposed AdaCorr is superior to several existing methods.

The following questions are expected to be addressed during rebuttal:
1.	The LRT-Correction procedure introduces additional computation costs. What is the difference in computation costs between Standard and the proposed label correction approach? Is the extra computation cost significant?

2.	The LRT-Correction procedure is introduced based on a binary setting. The main theorem (Theorem 1) also only supports the binary setting. There is a statement in Corollary 1 that “LRT-Correction can be generalized to multiclass classification tasks, by flipping \tilde{y} to be the best prediction of f when the null hypothesis is rejected. Theorem 1 can be generalized to multiclass classification tasks, by considering all pairs of class values.” How exactly did you do for that? Please provide more details.

3.	This paper uses the ablation study to demonstrate that the proposed AdaCorr is robust to several important hyper-parameters, e.g. the number of epochs m for the burn-in stage, the predefined value \Delta for likelihood ratio test. How did you exactly choose the optimal value for these hyper-parameters? These is a statement “We choose m=20 in this data set (CIFAR10) and “similarly” in other datasets.” Did you use the same m(=20) for all datasets? Does this also hold for the hyper-parameter \Delta ?

4.	The experiments are too standard. Any results on real-world datasets, e.g. Clothing1M [1]？

Minor comments:
1.	The summarization of the existing related work is not consistent throughout the paper. In Introduction section, this paper believes that the existing methods mainly follow two directions, i.e. probabilistic reasoning and data selecting; while in Related Work section, they are classified into three categories. Please clarify your arguments.

2.	Page 5: In Corollary 1, “LRT-Correctioncan” -> “LRT-Correction can”

3.	Page 7: In Table 2, MINIST -> MNIST.

4.	Page 7: In Experiment Setup, please provide more training details, e.g. learning rate.


[1] Xiao, Tong, Tian Xia, Yi Yang, Chang Huang, and Xiaogang Wang. "Learning from massive noisy labeled data for image classification." In Proceedings of the IEEE conference on computer vision and pattern recognition, pp. 2691-2699. 2015.



**Experience Assessment:**

I have published in this field for several years.

**Review Assessment: Checking Correctness Of Derivations And Theory:**

I assessed the sensibility of the derivations and theory.

**Review Assessment: Checking Correctness Of Experiments:**

I assessed the sensibility of the experiments.

**Review Assessment: Thoroughness In Paper Reading:**

I read the paper at least twice and used my best judgement in assessing the paper.

---

> ### Author Response · Authors · 2019-11-15
> **Reply to Reviewer #2**
>
> Thank you for the constructive comments. Below we address your concerns one-by-one.
>
> ** 1. Additional computation cost.
> A: For each training data, evaluating likelihood ratio involves comparing two posterior probabilities. This is true even for multi-class cases (see below for more details). The additional computation is O(1) for each training data at each iteration. In practice, the additional cost is marginal considering forwarding and backpropagation are the main bottlenecks. The retroactive loss is just like an other cross-entropy loss, except for it is comparing with the prediction of the model at a previous epoch.
>
> In experiments, our method takes less than twice longer than a base model (Standard) per iteration. On CIFAR10, to process each mini-batch (size 128), our algorithm takes 0.4 seconds while Standard takes 0.25 seconds. On Clothing 1M, for each mini-batch (size 32), our algorithm takes 0.33 seconds whereas Standard takes 0.29 seconds. We use nvdia GTX 1080TI GPUs.
>
> ** 2. Multi-class algorithm.
> A: Even for the multi-class setting, the label flipping always happens between two classes, the noisy label \tilde{y} and the current model prediction y*. The likelihood ratio is the ratio of their corresponding posterior probabilities, f(y=\tilde{y}|x) / f(y=y*|x). The denominator is the maximal posterior probability over all classes. When this ratio is smaller than \Delta, our model's best prediction is much stronger than its prediction w.r.t. the noisy label. Therefore, we flip the label to the model prediction, y*. Theorem 1 can be naturally generalized to the multi-class setting as it is only focusing on comparing the two classes, \tilde{y} and y*.  We have added the explanation and the multi-class algorithm into the revised paper (Procedure 2).
>
> ** 3. Hyperparameters m and \Delta.
> A: We have hyperparameters that need to be empirically decided, just like many other existing methods. For example, Co-teaching (Han et al., 2018) needs to choose the hyperparameter \tau properly to control the error flow of two networks; MentorNet (Jiang et al., 2018) similarly has to determine the hyperparameters \lambda_1 and \lambda_2 for regulating the learning pace; Forward (Patrini et al., 2017) needs to choose the number of epochs for pre-training the network in order to estimate the noise transition matrix.
>
> Our two hyperparameters, m and \Delta have intuitive/theoretical meanings. They are very easy to tune and are very robust. m is the well-known phase transitioning time between fitting common pattern and memorizing noise (Arpit et.al., ICML 2017). m can be selected by studying the training loss curve of the Standard model; it corresponds to the epoch when the training loss (on noisy label training set) slows down in its decreasing and starts converging. For MNIST, we simply chose m=15 for all noise patterns. For CIFAR10, we chose m=25 for all noise patterns.
>
> \Delta is the threshold in the LR testing. The theory provides guidance on how to decide \Delta. Indeed, we used a same strategy/setting for \Delta over all datasets and all noise patterns. First, \Delta is at most 1; if LR is bigger than 1, flipping is unnecessary as the noisy label \tilde{y} is the best prediction of the model. We chose an initial \Delta to be close to 1, in particular, 1/1.2. Then we slowly decrease \Delta as training continues. At the later training stage, with smaller \Delta, the LR testing is less likely to reject the hypothesis, and the flipping is less likely to happen. In other words, we are more and more conservative in flipping labels as training progresses. This provides a safety mechanism and ensures the choice of initial \Delta to be less sensitive.
>
> ** 4. Clothing 1M dataset.
> A: For clothing 1M dataset, we use pretrained resnet-50 and trained the model using SGD for 20 epochs. Our method has 71.47 accuracy. It outperforms Standard (68.94) and Forward (69.84). All other baselines (Forget, Decouple, MentorNet, Coteach) did not report result on this dataset.
>
> ** Thanks for pointing out these minor mistakes. We fixed them accordingly in the revised version.
>
> ** We use 1e-3 as the learning rate across all datasets.

---

### Author Response · Authors · 2019-11-15
**Summary of Revision**

We thank reviewers' constructive feedback. We revised our manuscript accordingly. Here is a list of changes.

** We provided the algorithm for multiclass classification setting (Procedure 2).

** We revised Theorem 1 to remove unnecessary linear assumption on the classifier f.

** We replaced O(\epsilon) with the actual constant in Theorem 1.

** In Section 3, we reported experimental results on Clothing 1M.

** In Section 4, we added experimental results on a clean dataset (Table 5) and convergence curve (Figure 3). We also discussed these results.

** We fixed typos and small presentation issues according to Reviewers' suggestions.

---

### Decision · Program_Chairs · 2019-12-19

**Decision:**

Reject

**Comment:**

This paper addresses a very interesting topic, and the authors clarified various issues raised by the reviewers.
However, given the high competition of ICLR2020, this paper is unfortunately still below the bar.
We hope that the detailed comments from the reviewers help you improve the paper for potential future submission.